# Antioxidant and Anticancer Functions of Protein Acyltransferase DHHC3

**DOI:** 10.3390/antiox11050960

**Published:** 2022-05-12

**Authors:** Chandan Sharma, Martin E. Hemler

**Affiliations:** Department of Cancer Immunology and Virology, Dana-Farber Cancer Institute, Boston, MA 02215, USA; chandan_sharma@dfci.harvard.edu

**Keywords:** oxidative stress, DHHC3, antioxidants, palmitoylation, cancer

## Abstract

Silencing of DHHC3, an acyltransferase enzyme in the DHHC family, extensively upregulates oxidative stress (OS). Substrates for DHHC3-mediated palmitoylation include several antioxidant proteins and many other redox regulatory proteins. This helps to explain why DHHC3 ablation upregulates OS. DHHC3 also plays a key role in cancer. DHHC3 ablation leads to diminished xenograft growth of multiple cancer cell types, along with diminished metastasis. Furthermore, DHHC3 protein is upregulated on malignant/metastatic cancer samples, and upregulated gene expression correlates with diminished patient survival in several human cancers. Decreased primary tumor growth due to DHHC3 ablation may be partly explained by an elevated OS → senescence → innate immune cell recruitment mechanism. Elevated OS due to DHHC3 ablation may also contribute to adaptive anticancer immunity and impair tumor metastasis. In addition, DHHC3 ablation disrupts antioxidant protection mechanisms, thus enhancing the efficacy of OS-inducing anticancer drugs. A major focus has thus far been on OS regulation by DHHC3. However, remaining to be studied are multiple DHHC3 substrates that may affect tumor behavior independent of OS. Nonetheless, the currently established properties of DHHC3 make it an attractive candidate for therapeutic targeting in situations in which antioxidant protections need to be downmodulated, and also in cancer.

## 1. Background

The cellular accumulation of excess reactive oxygen species (ROS) leads to oxidative stress (OS), which plays a key role in many types of human disease pathologies, including cancer [1,2,3]. Strategies to regulate excess oxidative stress for therapeutic benefit include activation of transcription factors for key redox-regulatory genes; stimulation of enzymes that remove excess ROS, such as superoxide dismutase (SOD), glutathione peroxidases (GPX), and catalase; and inhibition of enzymes such as NADPH oxidases (NOXs), which produce ROS [1]. However, in some cases, it may be more desirable to diminish antioxidant protection mechanisms. For example, in the case of circulating and potentially metastatic tumor cells, the loss of antioxidant protection allows excess ROS to push tumor cells beyond their toxic threshold [4,5,6]. Here we focus on the Golgi-resident enzyme DHHC3 as a novel type of oxidative stress regulator and a potentially useful drug target.

DHHC3 (also called GODZ) is one of at least 23 mammalian DHHC-type protein acyl transferases (PATs) identified so far in eukaryotes [7,8,9]. At least 12% of the mammalian proteome may be S-acylated, typically by palmitate [10], with nearly all S-acylation likely mediated by DHHC-type enzymes [9]. DHHC enzymes, localized in Golgi, endoplasmic reticulum (ER), or the plasma membrane, each contain a conserved “DHHC” (Asp–His–His–Cys) motif, within a larger conserved cysteine-rich domain of ~50 residues [11]. Two zinc atoms that are coordinated by conserved Cys residues stabilize the catalytic domain. The structures obtained for DHHC15 and DHHC20 [12] provide information supporting a two-step mechanistic model (Figure 1). In step one, palmitate is autocatalytically transferred from palmitoyl Coenzyme A (CoA) to an active site Cys residue (within the DHHC motif), where it protrudes into a cavity formed by four conserved DHHC transmembrane domains. In step two, palmitate is then transferred from the enzyme cavity directly to a substrate Cys residue. Although DHHC enzymes typically carry out S-acylation with palmitate, other acyl chains (e.g., myristate and stearate) can also be utilized, with some DHHC enzymes showing more variability in this regard [11].

The proximity of the DHHC active site to the inner membrane helps to explain why membrane-proximal cysteines are preferred substrate targets. Otherwise, rules governing DHHC enzyme substrate specificity are unresolved. In a particularly illustrative case, DHHC3 palmitoylates the integrin α6 but not α3 subunit, despite these two proteins being closely related overall and having similar amino acids flanking the site of palmitoylation [13]. Outside of the canonical four transmembrane (TM) domains and conserved cysteine-rich domain, DHHC enzymes show considerable variability [11,14], which undoubtedly contributes to substrate specificity in ways that remain to be defined.

Protein S-acylation can be reversed by depalmitoylating enzymes. Acyl-protein thioesterases (APT1 and APT2) may contribute more to depalmitoylation of peripheral membrane proteins, whereas the Alpha/Beta Hydrolase Domain-containing Protein 17 (ABHD17) family of hydrolases may depalmitoylate proteins more proximal to the plasma membrane [15].

Among the DHHC-type enzymes, DHHC3 has emerged as a key regulator of OS, tumor growth, and metastasis. Hence, it is the focus of this review.

## 2. DHHC3 and Oxidative Stress Regulation

### 2.1. DHHC3 Ablation Enhances OS

Among the DHHC-type enzymes, the regulation of OS has not been a major theme. The presence of DHHC1 was shown to induce OS [16], whereas the absence of DHHC13 caused mitochondrial dysfunction accompanied by elevated OS [17]. Now, DHHC3 has emerged as a major regulator of OS. Multiple cancer cell lines displayed elevated oxidative stress following RNAi mediated ablation of DHHC3. Evidence for elevated oxidative stress includes enhanced CellRox dye fluorescence and the upregulation of oxidative stress marker TXNIP [18,19]. TXNIP (thioredoxin inhibitory protein) supports oxidative stress by inhibiting the function of TRX-1 (thioredoxin protein-1) [20,21]. DHHC3 ablation also resulted in elevated tyrosine phosphorylation of select kinases (Focal Adhesion Kinase (FAK) and Signal Transducer and Activator of Transcription 3 (STAT3)), and this is consistent with OS-induced loss of phosphatase activity [18]. Either the ablation of TXNIP or addition of OS inhibitors NAC (N-acetyl cysteine), α-LA (alpha-lipoic acid), and atorvastatin substantially prevented OS-inducing effects of ZDHHC3 ablation [18].

An analysis of DNA array results provided further supporting evidence for OS regulation. Specifically, out of the 52 genes with significantly altered expression due to DHHC3 ablation, six upregulated genes (*GTF2i*, *TXNIP*, *AVIL*, *FKBP11*, *SETD6*, and *SETX*) and six other downregulated genes (*S100A4*, *PDE4B*, *HNMT*, *NUDT2*, *AKR1C1*, and *GSTZ1*) showed changes in accordance with results expected from elevated oxidative stress [18].

### 2.2. DHHC3 Substrates Include Several Oxidative Stress Regulators

DHHC3 mutated within its enzyme active site failed to exert antioxidant functions, thus affirming the importance of its palmitoylation activity [18,19]. To gain insight into the mechanism of oxidative stress regulation by DHHC3, protein substrates palmitoylated by DHHC3 were identified by using a PalmPISC (palmitoyl protein identification and site characterization) approach, in which global palmitoyl-proteomic methodology and mass spectrometric identification were coupled with stable isotope labeling by amino acids in cell culture (SILAC). This approach resulted in the first comprehensive and unbiased identification of DHHC3 substrates from both breast and prostate cancer cell lines [19]. Among the different substrates, 22–28 antioxidant/redox-regulatory proteins were identified, including several that were common to both cell types. The use of unbiased Ingenuity Pathway Analysis (IPA) linked both sets of candidate substrates primarily to oxidative stress, free radical scavenging, and mitochondrial dysfunction, which causes oxidative stress [22]. DHHC3 substrates present in one or both breast and prostate cell lines included notable antioxidant proteins such as Glutathione Peroxidase 8 (GPX8) [23], Thioredoxin Related Transmembrane Protein 3 (TMX3) [24], Peroxiredoxin 1 (PRDX1) [25], Peroxiredoxin 4 (PRDX4) [26], and Peroxiredoxin 5 (PRDX5) [27]. In addition, multiple other substrates were identified which were functionally linked to oxidative stress and/or endoplasmic reticulum-stress [19]

These results are central to understanding the antioxidant functions of DHHC3. When protein palmitoylation is prevented, affected proteins typically show altered subcellular localization and/or enhanced degradation [14,28]. Indeed, we confirmed for several DHHC3 substrates (including Endoplasmic Reticulum-Golgi Intermediate Compartment protein 3 (ERGIC3), NPC Intracellular Cholesterol Transporter 1 (NPC1), Transmembrane Protein 192 (TMEM192), and PRDX4) that the loss of palmitoylation, due to DHHC3 ablation, is accompanied by markedly altered subcellular distribution [18,19], which predicts diminished function. It was established that ERGIC3 ablation mimicked pro-OS effects of DHHC3 ablation, and this is consistent with ERGIC3 being a key substrate [18]. However, experiments (e.g., involving the mutation of key palmitoylation sites) still remain to be performed to determine which DHHC3 substrates are most critical for DHHC3’s antioxidant function. In the absence of additional data, our working hypothesis is that the simultaneous disruption of palmitoylation and function of several redox regulatory proteins likely explains elevated oxidative stress in DHHC3-ablated cells [18,19]. The results relevant to OS regulation by DHHC3 are listed in Table 1. The role of DHHC3 in the palmitoylation and function of key antioxidant protein substrates is summarized in Figure 2.

## 3. DHHC3 and Cancer

Expression levels for several DHHC enzymes have been linked to cancer-related events [14], but detailed mechanistic insight has been lacking. Conversely, several proteins with oncogenic or tumor-suppressor properties require S-acylation to function properly [14], but, in many cases, the relevant DHHC enzyme(s) has/have not been identified. However, in one notable example, the melanocortin-1 receptor (MC1R) requires palmitoylation by DHHC13 to enable its tumor-suppressor function in melanocytes [29]. In the case of DHHC3, the combination of genetic ablation studies and the comprehensive identification of many substrates in multiple tumor cell lines now allows a more complete story to emerge.

Elevated *ZDHHC3* gene expression correlates with diminished patient survival in at least seven different cancer types [18]. DHHC3 protein was significantly upregulated in malignant human breast cancer, and to an even greater extent in metastatic breast cancer samples [18], in addition to prostate and colon cancers [30]. In both orthotopic and ectopic mouse models of breast cancer, primary tumor growth from injected tumor cells was significantly reduced upon DHHC3 ablation [18]. Primary tumor growth was also significantly reduced in ectopic models of prostate [19] and colon [31] cancers. Tumor cells reconstituted with wild-type DHHC3, but not palmitoylation site-mutated DHHC3, regained tumor growth, thus confirming the importance of DHHC3-mediated palmitoylation. Upon DHHC3 ablation, tumor metastasis was also significantly reduced, both in colony size and number, in a mouse-tail-vein injection model. Together, these results, which are summarized in Table 2, indicate a key role for DHHC3 in cancer. In general, lower levels of ROS are pro-tumorigenic, whereas high ROS levels are cytotoxic [32]. Hence, several of the effects listed in Table 2 likely involve the antioxidant activity of DHHC3.

## 4. DHHC3 Impact on Innate Anticancer Immunity

Reduced tumor growth by DHHC3-ablated breast tumor cells may be at least partly due to OS-triggered senescence [33,34], leading to innate anticancer immunity [18]. In this regard, DHHC3 ablation resulted in induction of premature senescence in multiple cell types, as evidenced by the upregulation of β-galactosidase activity. Moreover, there was secretion of a specific set of chemokines that are markers of SASP (senescence associated secretory phenotype) [35]. Additionally, gene microarray analyses of the xenograft tumors demonstrated the upregulation and downregulation of multiple senescence-linked genes [18]. Confirming the central role of OS, these effects of DHHC3 ablation on senescence induction were reversed by treatment of cells with oxidative stress inhibitors such as NAC, α-LA, and atorvastatin, along with redox protein (TRX) inhibitor, TXNIP. Consistent with a SASP response (37), innate immune cells (antitumor “M1-like” macrophages and NK cells) were recruited into breast xenograft tumor sites. Furthermore, conditioned media from ZDHHC3-ablated cells (containing SASP chemokines MCP-1 and IL8) showed higher recruitment of antitumor “M1-like” macrophages in an in vitro model system [18]. Together, this evidence strongly suggests that silencing of DHHC3 upregulates OS, which plays a key role in senescence induction and subsequent recruitment of innate immune cells into the tumor microenvironment, followed by tumor inhibition.

Consistent with this finding, other studies have also shown that the manipulation of either oncogene, tumor suppressor, or other stress-inducing genes can induce premature senescence, leading to immune-cell-mediated tumor clearance. For instance, in hepatocellular cancer, the re-expression of p53 (tumor suppressor) promoted immune clearance of carcinoma cells by senescence induction during in vivo studies [36,37] Similarly, oncogene BRAF also induced senescence and suppressed tumor xenograft growth [38,39], and the involvement of the cGAS-STING pathway was seen in oxidative-stress-induced senescence [40]. Consistent with the potential therapeutic efficacy of senescence induction, as suggested by these studies, the effects of Cyclin-Dependent Kinases 4 and 6 (CDK4/6) specific inhibitor Ribociclib (LEE011) which induced cell-cycle arrest and senescence in human neuroblastoma, is being investigated in clinical trials [41,42].

However, in contrast to the abovementioned studies, the pro-inflammatory effects of SASP, acting through interleukin-dependent networks, have also been observed to support tumor growth [43]. This serves as a reminder that tumor-suppressive effects of senescence are context-dependent. Nonetheless, in some model systems, DHHC3 silencing does appear to involve upregulated OS, leading to the senescence and recruitment of innate immune cells, resulting in diminished tumor growth.

## 5. DHHC3 Mediated Regulation of Adaptive Anticancer Immunity

Silencing of DHHC3 in a colon carcinoma cell line has been shown to enhance adaptive immunity [31]. Because ROS can support the activation, differentiation, and survival of T and B cells [44,45], and DHHC3 ablation induces OS [18,19], this could at least partly explain enhanced adaptive immunity. Alternatively, enhanced adaptive immunity could involve diminished Programmed Death-Ligand 1 (PD-L1) on the tumor cells. (CKLF like MARVEL Transmembrane Domain Containing 6 (CMTM6) [19], which indirectly supports expression of immune checkpoint inhibitor PD-L1 [46,47], is palmitoylated by DHHC3 [19]. Hence, loss of CMTM6 function due to DHHC3 ablation should lead to diminished PD-L1, resulting in enhanced adaptive antitumor immunity.

It has also been suggested that DHHC3 may directly palmitoylate PD-L1, thus explaining why DHHC3 silencing leads to enhanced adaptive immunity [31]. However, a conflicting report suggests that PD-L1 is palmitoylated instead by DHHC9 [48]. Furthermore, we have not observed PD-L1 to be palmitoylated [19], and according to the Swiss Palm database, PD-L1 (CD247) has not yet been found as a candidate in any S-palmitoylation proteomic studies of various organisms, including mammals.

ROS affects adaptive immunity both positively and negatively by various mechanisms involving several different cell types [32,44,45]. For example, high levels of ROS can promote immunosuppression rather than enhance adaptive immunity [32,44,45]. Thus, it is possible that DHHC3 ablation could sometimes inhibit adaptive immunity. At present, more experimental evidence is needed, using multiple tumor models, and perhaps other disease models, to properly determine the parameters of DHHC3-dependent regulation of adaptive immunity.

## 6. DHHC3 Regulation of Metastasis

A Severe Combined Immunodeficiency (SCID) mouse-tail-vein injection model displayed markedly diminished tumor metastasis upon DHHC3 ablation [18]. Since adaptive anticancer immunity is absent, and innate anticancer immunity is minimal in this model, these results suggest an immune-independent mechanism. In this regard, circulating tumor cells and tumor metastasizing cells have been shown to be selectively sensitive to elevated oxidative stress [6] and can be made more sensitive by the disabling of antioxidant protections [4,32,49]. As DHHC3 ablation alone is sufficient to elevate oxidative stress, we predict that immune-independent mechanisms may be sufficient to explain reduced metastasis by DHHC3-ablated cells; however, definitive experiments are still needed.

## 7. DHHC3 Mediated OS and Anticancer Drug Efficacy

### 7.1. Anticancer Drugs and Oxidative Stress

Chemotherapeutic agents typically trigger the elevation of intracellular oxidative stress, which is a critical aspect of drug lethality. Conversely, the potency of some chemotherapeutic agents is attenuated (chemoresistance) by the upregulation of antioxidant expression through a feedback loop mechanism, which helps to protect cells by countering excess oxidative stress [4,5,50]. A few other classes of drugs, e.g., Poly-ADP Ribose Polymerase (PARP) inhibitor PJ-34, may also trigger oxidative stress, with the lethal effects again being reversed upon the addition of an antioxidant [51]. Consequently, to enhance drug efficacy and overcome chemoresistance, attempts are being made to increase ROS and/or disable antioxidant protection mechanisms [4,5,50].

### 7.2. DHHC3 Disruption Enhances OS-Inducing Drug Effects

Because DHHC3 can control OS by simultaneously supporting the expression and function of multiple antioxidant proteins, we predicted that the ablation of DHHC3 should diminish protective antioxidant responses to OS-inducing drugs. Indeed, the combination of DHHC3 ablation with OS-inducing anticancer drugs resulted in synergistically enhanced oxidative stress. This synergy was also manifested as increased apoptosis, diminished cell proliferation, and/or enhanced cell cycle disruption [19]. Notably, reconstitution with functional DHHC3 restored protection from drug-induced oxidative stress and apoptosis, whereas palmitoylation-deficient (active site mutant) DHHC3 did not. These results firmly point to the importance of DHHC3-mediated palmitoylation of redox regulatory substrates, while ruling out the off-target effects of DHHC3 ablation. Furthermore, it is strongly suggested that DHHC3 ablation/inhibition may be a novel means to overcome chemoresistance by simultaneously disabling multiple antioxidant protection mechanisms while upregulating oxidative stress.

## 8. Other DHHC3 Substrates May Affect Cancer Independent of OS

Unbiased analysis revealed that redox regulation is a major theme among DHHC3 substates [19], and OS arising from DHHC3 ablation may affect tumor progression at multiple levels (see above). Nonetheless, there are several other prominent DHHC3 substrates (i.e., near the top of the substrate list with respect to DHHC3-dependent palmitoylation [19]) that could affect tumor cell behavior independent of OS. For example, Nucleolar Protein 6 (NOL6) may have oncogenic functions in endometrial [52] and prostate [53] cancers, while CMTM3 may contribute to proliferation, migration, and poor survival in prostate cancer [54]. Moreover, Chromobox Protein Homolog 5 (CBX5) is elevated in colorectal cancer [55] and supports tumor stem-like properties in lung cancer [56], and SUMF2 is highly mutated in colorectal cancer [57]. The integrin α6 subunit (ITGA6) makes major contributions to colorectal cancer [58], hepatocellular cancer [59], and several other cancers, while also supporting stem-cell function in many cancers [60]. We hypothesize that proper palmitoylation by DHHC3 is needed for pro-cancer functions of the proteins mentioned here (and schematically in Figure 2), but this remains to be experimentally demonstrated.

## 9. DHHC3 as a Potential Drug Target

### 9.1. Reasons Why DHHC3 May Be an Excellent Target for Therapeutic Intervention

First, its removal caused diminished tumor growth in three different xenograft models [18,19,31]. These results are consistent with upregulated *ZDHHC3* gene expression being correlated with poor cancer patient survival [18]. Second, its removal caused reduced metastasis [18], likely by an immune-independent mechanism involving elevated oxidative stress. Third, elevated OS, caused by DHHC3 silencing, may trigger innate anticancer immunity [18]. Fourth, DHHC3 silencing appeared to enhance adaptive anticancer immunity, possibly by mechanisms involving regulation of PD-L1 expression on tumor cells [19,31]. Fifth, DHHC3 ablation disables antioxidant protection mechanisms, due to the simultaneous disruption of several key redox regulatory proteins [19]. Hence, a newly developed DHHC3 inhibitor would be a useful tool for enhancing OS when appropriate. For example, as seen for DHHC3 ablation [19], a selective DHHC3 inhibitor should also enhance the lethality of OS-inducing anticancer drugs. Sixth, DHHC3 expression is selectively upregulated on malignant/metastatic breast cancer cells [18] and other cancer cell types [30]. Hence, those cells should be more sensitive to DHHC3 targeting, whereas redox balance would perhaps be less disrupted in normal cells that have lower DHHC3 levels. Seventh, absence of DHHC3 from mice causes minimal overall effects on animal physiology [61], thus predicting that DHHC3 targeting should have minimal side effects.

### 9.2. Implications of Targeting DHHC3

In support of the studies mentioned here, DHHC3 was also recently reviewed elsewhere as a potentially novel antitumor target [62]. Regarding the fourth and fifth points mentioned above, a recent clinical trial found it beneficial for cancer patient survival to combine anti-PD-L1 therapy with chemotherapy [63]. We suggest that the inhibition of DHHC3 would likely augment both anti-PD-L1 therapy and chemotherapy at the same time. However, due to the many circumstances in which antioxidant protection is beneficial [1], caution would be required when inhibiting DHHC3. Furthermore, the wide variety of DHHC3 substrates (both redox-related and unrelated [19]) introduces additional unpredictability with regard to DHHC3 inhibition.

## 10. Conclusions

Recent discoveries indicate that DHHC3 has strong antioxidant properties. This helps to explain why DHHC3-ablated cells show elevated OS in concert with diminished tumor growth and metastasis, and increased sensitivity to OS-inducing drugs. DHHC3 likely also regulates cancer by OS-independent mechanisms, but this remains to be confirmed. An abundance of evidence strongly supports DHHC3 as a novel cancer therapeutic target. However, effective and specific drugs targeting DHHC-type enzymes have not yet emerged. Nonetheless, efforts are being made toward targeting DHHC-type enzymes, both in cellular and cell-free contexts [64,65,66], and a candidate inhibitory peptide has been identified [31]. Recent determinations of crystal structures for human DHHC20 and zebrafish DHHC15 [12] represent a key step forward, which should enhance future efforts at drug discovery [11].

## Figures and Tables

**Figure 1 antioxidants-11-00960-f001:**
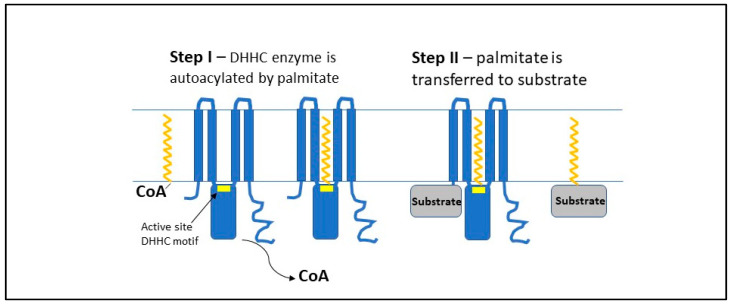
**Proposed mechanism for palmitoylation catalyzed by DHHC enzymes.** Step I is an autoacylation step, in which palmitate is transferred from palmitoyl–CoA thioester to the active site cysteine within the DHHC motif conserved among all DHHC enzymes [7,8,9]. Based on structural data obtained for hDHHC20 and zfDHHC15 [12], the DHHC motif (yellow block within the blue enzyme) is located at the membrane–cytoplasmic interface, and the attached palmitate transiently resides within the membrane lipid bilayer in a cavity formed by four canonical DHHC transmembrane domains. Step II involves transfer of palmitate from the DHHC–thioester directly to a membrane-proximal cysteine of a substrate protein.

**Figure 2 antioxidants-11-00960-f002:**
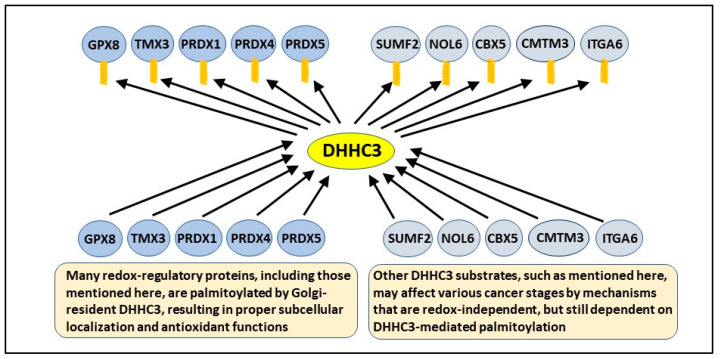
**Key substrates most likely responsible for DHHC3 functions.** During passage through the Golgi, several newly synthesized proteins are palmitoylated by DHHC3. For several antioxidant proteins (e.g., GPX8, TMX3, PRDX1, PRDX4, and PRDS5), palmitoylation facilitates their proper localization and function, thus controlling oxidative stress. Similarly, DHHC3-mediated palmitoylation enables the functions of additional proteins (e.g., SUMF2, NOL6, CBX5, CMTM3, and ITGA6) that may support various stages of tumorigenesis independent of redox regulation.

**Table 1 antioxidants-11-00960-t001:** Consequences of DHHC3 ablation consistent with elevated OS.

Result	References
Enhanced CellRox fluorescent dye detection	[18,19]
Diminished activity of select tyrosine phosphatases	[18]
Effects of DHHC3 ablation partially reversed by OS inhibitors	[18]
Upregulated appearance of TXNIP	[18]
Changes in gene expression consistent with elevated OS	[18]
Diminished palmitoylation of Redox/Antioxidant regulators	[19]
Increased senescence, innate immune cells in tumors	[18]
Increased efficacy of OS-inducing anticancer drugs	[19]

**Table 2 antioxidants-11-00960-t002:** DHHC3 plays a key role in cancer.

Result	References
Upregulated *ZDHHC3* gene levels and diminished patient survival	[18]
DHHC3 upregulated in breast, prostate, and colon cancers	[18,30]
Xenograft tumor growth reduced upon DHHC3 ablation	[18,19,31]
Tumor metastasis is reduced for DHHC3-ablated cells	[18]

## Data Availability

Data relevant to this manuscript are contained in [18,19].

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
