# Peer review of "Antioxidant and Anticancer Functions of Protein Acyltransferase DHHC3"

_antioxidants, 2022, doi:10.3390/antiox11050960_

Round 1

Reviewer 1 Report

Dear authors:

The manuscript entitled ‘Antioxidant and Anti-cancer Functions of Protein Acyltransferase DHHC3’ mainly reviewed the functions and roles of DHHC3 in antioxidant and anti-cancer. It is an original article and catches some parts to be adjusted which are as follows:

  1. I think the organization of this manuscript should be improved.
  2. Mechanisms in this manuscript could be more creative and the main emphasis should be highlighted.
  3. Review is not the simple list of articles, the authors should have your own views and speculations, such as to summarize the common mechanisms of DHHC3 in antioxidant and anti-cancer and sum up the function.
  4. A more detailed illustration could be made to show the mechanism of DHHC3.
  5. Finally, the abstract was templating, and the authors should make the language more native by appropriate modifications.

Thank the authors for providing us your painstaking effort.

Thank you and best regards.

Yours sincerely

Reviewer 2 Report

The authors present a review of antioxidant and anti-cancer functions of protein acyltransferase DHHC3. They have done an excellent job summarizing the current state of the art. The key references are here and the work is set out logically. The only critiques I have are more stylistic and can be dealt with at the proof stage.

- The font of figure 1 could be more professional and the blue boxs removed.

- Section 8 could be sub-headed and cleaned up.

Author Response

Please see the attachment (which contains all three reviewer responses.

Reviewer 3 Report

In this manuscript by Sharma et al, the authors summarized recent findings of DHH3C, a protein acyltransferase whose down-regulation is closely related to increased oxidative stress and cell death, which can be a promising target for cancer therapy. While this review may draw interest in a specific field of cancer research, there are a number of concerns that need to be addressed before this manuscript is in a publishable fashion. Specific comments are as follows:

1. This review is based heavily on the two original articles of DHH3C in cancer research from the same laboratory. Although both are elegant studies, this manuscript does not really add more information about this specific area of research.

2. The authors discussed the DHH3C substrates related to oxidative stress and the effects of senescence on tumor immunity, but their association with DHH3C biological activity was not well established or not scientifically proven. The true cellular properties of DHH3C may be misled. 

3. An introduction of DHHC family of proteins is highly suggested.

Author Response

Please see the attachment (which contains all three reviewer responses).

Round 2

Reviewer 3 Report

The revised version by Sharma et al. has tremendous improvement with the information of the DHHC family and added sections. The reviewer has no further questions.